# Enhancing Digestibility of *Chlorella vulgaris* Biomass in Monogastric Diets: Strategies and Insights

**DOI:** 10.3390/ani13061017

**Published:** 2023-03-10

**Authors:** Maria P. Spínola, Mónica M. Costa, José A. M. Prates

**Affiliations:** 1CIISA-Centro de Investigação Interdisciplinar em Sanidade Animal, Faculdade de Medicina Veterinária, Universidade de Lisboa, 1300-477 Lisboa, Portugal; 2Laboratório Associado para Ciência Animal e Veterinária (AL4AnimalS), Faculdade de Medicina Veterinária, Universidade de Lisboa, 1300-477 Lisboa, Portugal

**Keywords:** microalgae, *Chlorella vulgaris*, bioavailability, bioaccessibility, digestibility, poultry, swine

## Abstract

**Simple Summary:**

This study aimed to review the potential of the microalgae *Chlorella vulgaris* (CV) as an animal feed source especially for monogastric animal diets, due to their high content of essential nutrients. The findings of a systematic literature review showed that adding CV to poultry and swine diets had different results in terms of nutrient digestibility, although pre-treatments increased nutrient accessibility and digestibility. Cost-effectively produced CV biomass has the potential to be a supplement or substitute for expensive feed ingredients and improve animal health and immunity. Variations in results may be due to differences in microalgal strain, cultivation conditions and dietary inclusion levels. This study provides new insights into the use of CV biomass in animal diets.

**Abstract:**

Microalgae, such as *Chlorella vulgaris* (CV), have been identified as promising animal feed sources due to their high content of essential nutrients, including proteins, total lipids, n-3 polyunsaturated fatty acids, and pigments. This study aimed to review the digestibility, bioaccessibility, and bioavailability of nutrients from CV biomass, and to analyse strategies to enhance their digestibility in monogastric animal diets. The study conducted a systematic review of the literature from databases such as PubMed, Scopus, Google Scholar, and Web of Science, up until the end of January 2023. The results of adding CV to poultry and swine diets were diverse and depended on a number of variables. However, pre-treatments applied to CV biomass improved nutrient digestibility and accessibility. CV biomass, produced in a cost-effective manner, has the potential to serve as a supplement or substitute for expensive feed ingredients and improve animal health, physiology, and immune status. Variations in results may be due to differences in microalgal strain, cultivation conditions, and dietary inclusion levels, among other factors. This study provides new insights and perspectives into the utilization of CV biomass in animal diets, highlighting its potential as a valuable ingredient to improve nutrient utilization.

## 1. Introduction of *Chlorella vulgaris*

Microalgae, such as *Chlorella vulgaris*, have garnered interest as food and feed sources due to their high growth rate and rich content of essential nutrients, particularly protein [1,2,3]. Unlike traditional food crops, microalgae can grow in a variety of environments without requiring arable land, and, therefore, avoid competition with crops, such as soybean and cereal grains [4]. In addition, the ability of microalgae to fix carbon from the atmosphere makes them a valuable tool in reducing greenhouse gas emissions [5]. Therefore, the reasons for choosing microalgae as an ingredient or supplement for animal feed are, for instance, their high nutritional value, if cultured under optimal conditions and fed to adapted animals [6], and their ability to act as feed additives. Particularly, extracted algal bioactive compounds can improve animal production by exerting antimicrobial and immunomodulatory activities [7] or inhibit methanogenesis in ruminants with positive effects on reducing methane emissions [8,9]. These factors have driven the exploitation of microalgae for human nutrition, animal feed, biofuels, and bioremediation in recent years [10]. For instance, fresh, dried, or frozen microalgae can be used as raw material, extracted oil, or algal extracts for supplementation of monogastric animal diets [2].

*Chlorella* is a genus of green algae (*Chlorophyceae*) that can grow in a variety of habitats, including seawater, freshwater, and soil. It can grow autotrophically, heterotrophically, or mixotrophically, and is usually cultured in raceway ponds or photobioreactors [11]. The nutritional composition of *Chlorella* is influenced by growing conditions and species, but it is a rich source of high-quality protein, minerals, vitamins, and pigments [5]. For example, *Chlorella vulgaris* (CV) can contain up to 67% dry matter (DM) of protein, with digestibility coefficients comparable to those of beans, oats, and wheat [12,13]. Additionally, CV is normally rich in lipids, with a high content of polyunsaturated fatty acids, and carbohydrates [6,14]. The nutritional properties of *Chlorella* have led to its use as an animal feed supplement or ingredient, with studies showing positive effects on growth performance, immune function, and meat quality in poultry and swine [11,15]. Despite its potential as a feed ingredient, there are limitations in the digestibility of CV biomass by monogastric animals [4]. This is due to the presence of cross-linked insoluble carbohydrates in its recalcitrant cell wall, including cellulose and chitin-like polymers [16,17]. The structure of the cell wall, which can consist of one or two microfibrillar layers, is dependent on the strain of CV and its stage of growth [16,18]. Efforts have been made to enhance digestibility by using mechanical and chemical methods [17,19,20], but further improvement is necessary to achieve optimal bioavailability and bioaccessibility for monogastric animals [21]. In addition, it is crucial to address the cost and environmental impact of microalgae production, including the use of recycled nutrients and water, as well as carbon sources from flue gases and anaerobic digestion products [5,11,22].

The present study systematically reviews the in vivo and in vitro digestibility, bioaccessibility, and bioavailability of nutrients present in CV biomass, and highlights the strategies to improve their digestibility in monogastric animal diets (Figure 1). Therefore, this review, which was conducted up until the end of January 2023, fills this gap of knowledge by covering the existing literature available in the reference databases of PubMed (NCBI, Bethesda, MD, USA), Scopus (Elsevier B.V., Netherlands), Google Scholar (Google LLC, Mountain View, CA, USA), and Web of Science (Clarivate Analytics, Philadelphia, PA, USA). The literature search was performed using the keywords “*Chlorella vulgaris*”, “*Chlorella*”, “bioavailability”, “bioaccessibility”, “digestibility”, “poultry”, “swine”, and “pig”. The ultimate aim was to identify ways to improve the digestibility, bioaccessibility, and bioavailability of nutrients from CV biomass.

## 2. Nutritional Composition of *Chlorella vulgaris*

The chemical composition of CV is rich in various nutrients including protein, which can reach levels of up to 65.5% dry matter (DM) [23]. However, its protein content is highly dependent on the cultivation conditions and can be as low as 13.6% DM [24] under nitrogen-limiting conditions [25]. The protein quality of CV is high, as it contains all essential amino acids, including a significant amount of leucine and lysine (average of 9 to 10% of total amino acids) [26,27]. Lysine, in particular, is a limiting amino acid for poultry and swine, and can be found in *C. vulgaris* at levels as high as 10.4 to 13.2% [26,27].

The dried biomass of CV also contains significant amounts of ash, reaching up to 27.3% [28]. The ash is rich in essential minerals such as phosphorus, potassium, iron, manganese, and zinc [24,26]. While there may be some concern about heavy metal toxicity in *Chlorella* spp., studies have shown that such toxic elements, including arsenic, cadmium, and mercury, are present in concentrations as low as 0.59–1.1 mg/kg, 0.01–0.10 mg/kg, and 0.02–0.10 mg/kg, respectively, and no lead was detected [25]. These low levels of heavy metals ensure the safety of CV as a dietary supplement.

CV contains a variable amount of carbohydrates, which can range from 8.08 [29] to 65.0% [23], with an average of 23.4% DM (Table 1). The cell wall of CV is composed of hemicellulose (22–25%) and chitin-like polysaccharides (60–66%) [16], and also contains a small amount of starch (up to 4.41% DM). The variability in carbohydrate content is largely influenced by growth conditions and the stage of algal growth [16,30]. In early stages of cell growth, the cell wall of CV is composed of a single microfibrillar layer, while in later stages, a two-layer structure appears with a thick outermost layer and a thinner inner layer separated by an electron translucent interspace [16].

The total lipid content of CV is significant, with an average of 12.1% DM. However, the lipid content can vary significantly, ranging from 5.10 [31] to 19.7% DM [32]. The variability is largely due to differences in nitrogen supply, as nitrogen limitation can increase lipid content while decreasing protein content [25]. Environmental stress, such as high light intensity and nitrogen deprivation, has also been shown to increase lipid production in other microalgae [33]. The essential PUFA 18:2n-6 and 18:3n-3 are the most predominant fatty acids in CV biomass, averaging 21.6% and 18.8%, respectively [23,26,32]. The 18:3n-3 and 18:2n-6 are converted into n-3 LC-PUFA and n-6 LC-PUFA, respectively, although the efficiency of these pathways is low [34]. Nevertheless, including CV in animal diets, such as piglet [15] and finishing pig [35], can increase the n-3 PUFA content in meat and improve its nutritional value.

CV is a rich source of pigments such as chlorophylls *a* and *b* and carotenoids (*e.g*., β-carotene and lutein) [20]. The total chlorophyll and carotenoid content in CV can vary and reach levels up to 24.0 [28] and 3.49 g/kg DM [28], respectively, depending on the drying process, culturing conditions, and harvest time [36]. These pigments have significant antioxidant and radical scavenging properties [36]. The accumulation of carotenoids in the *longissimus lumborum* muscle of finishing pigs was shown to enhance the nutritional value of meat without affecting its colour [35]. In addition, the accumulation of carotenoids in egg yolks [37] and broiler chicken meat [38] has been reported to cause a decrease in redness (a*) or an increase in yellowness (b*) parameters, which could potentially impact consumer perception of the egg or meat [6]. Additionally, CV is a valuable source of vitamin E (α-tocopherol) [35] and B-complex vitamins, with a prevalence of niacin (vitamin B3), with levels ranging from 145 [31] to 247 [39] mg/kg DM, as well as active forms of vitamin B12 (cyanocobalamin) [40].

**Table 1 animals-13-01017-t001:** Chemical composition of *Chlorella vulgaris* (all values are expressed on a dry matter basis; hyphenated values are ranges based on several studies and mean values are within parenthesis).

Nutritional Composition	*Chlorella vulgaris* ^1^
Crude protein (%)	13.6–65.5 (41.7)
Amino acid profile (% total amino acids)	
Alanine	6.95–10.9 (8.27)
Arginine	6.68–14.2 (9.33)
Aspartic acid	6.84–10.9 (9.18)
Cystine/Cysteine	0.01–1.81 (0.55)
Glutamic acid	9.08–13.4 (11.1)
Glycine	4.67–8.60 (6.66)
Histidine	1.26–4.12 (2.14)
Isoleucine	0.10–6.58 (3.59)
Leucine	6.65–19.5 (9.87)
Lysine	6.84–13.2 (9.46)
Methionine	0.65–2.67 (1.70)
Phenylalanine	4.05–11.3 (6.17)
Proline	2.97–5.29 (4.66)
Serine	4.24–7.78 (5.21)
Threonine	5.06–10.9 (6.50)
Tryptophan	0.003–3.09 (1.65)
Tyrosine	3.20–8.44 (5.07)
Valine	3.09–14.4 (7.51)
Crude carbohydrates (%)	8.08–65.0 (23.4)
Non-fibre carbohydrates (starch)	2.00–4.41 (3.20)
Crude fibre (%)	1.63–5.98 (3.81)
Acid detergent fibre	0.31–9.78 (4.64)
Neutral detergent fibre	0.18–16.4 (5.90)
Crude fat (%)	5.10–19.7 (12.1)
Fatty acid profile (% total fatty acids)	
16:0	15.4–29.1 (21.2)
16:1n-7	0.35–2.90 (1.14) ^2^
16:1n-9	3.59–3.90 (3.75)
18:0	0.72–6.50 (3.96)
18:1n-9	2.10–33.1 (11.7)
18:2n-6	8.37–40.3 (21.6)
18:3n-3	1.93–34.8 (18.8)
18:3n-6	0.04–4.45 (2.46)
20:0	0.03–0.25 (0.17)
20:4n-6	0.13–0.98 (0.57)
20:5n-3	0.19–3.23 (0.74)^3^
22:5n-6	0.10–2.00 (0.94)
22:6n-3	0.15–20.9 (2.32)^3^
24:0	0.35–4.00 (2.92)
Ash (%)	6.30–27.3 (11.8)
Macrominerals (g/kg)	
Calcium	0.36–53.3 (10.1)
Magnesium	0.41–16.4 (7.02)
Phosphorus	5.11–27.1 (17.1)
Potassium	0.50–133 (32.2)
Sodium	0.50–16.5 (8.80)
Microminerals (mg/kg)	
Copper	0.00–31.1 (14.2)
Iron	190–5400 (1450)
Manganese	20.9–1270 (328)
Selenium	0.17–0.70 (0.44)
Zinc	11.9–530 (165)
Pigments (g/kg)	
Total carotenoids	0.24–8.21 (3.49)
β-carotene	0.007–1.88 (0.70)
Lutein	0.05–0.87 (0.46)
Total chlorophylls	1.16–24.0 (10.9)
Chlorophyll a	0.50–18.3 (7.00)
Chlorophyll b	0.07–5.65 (2.05)
Vitamins (mg/kg)	
A	22.6
B1	6.74–15.6 (11.2)
B2	28.0–49.8 (40.8)
B3	145–247 (211)
B5	13.5
B6	15.2–17.6 (16.4)
B7	1.99
B9	0.28–19.9 (10.1)
B12	0.26–2.29 (0.91)
C	162
K	122
α-Tocopherol	20.6
β- Tocopherol	0.37
γ- Tocopherol	0.56

^1^ Supporting sources: Canelli et al. [23]; Sucu [24]; Cabrita et al. [26]; Shaaban [27]; Madhubalaji et al. [28]; Tokuşoglu and üUnal [29]; Prabakaran et al. [31]; Ferreira et al. [32]; Coelho et al. [35]; Panahi et al. [39]; Edelmann et al. [40]; Gonzalez and Bashan [41]; Janczyk et al. [42]; Khoeyi et al. [43]; Batista et al. [44]; Safi et al. [45]; Kholif et al. [46]; Jalilian et al. [47]. ^2^ Contains trace amounts of other 16:1 isomers. ^3^ Below limit of detection in one study.

## 3. Enhancing the Digestibility of *Chlorella vulgaris* Nutrients

Disruption of algal cell wall is necessary to improve the digestibility and bioaccessibility of CV [21]. This can be achieved through mechanical/physical procedures, such as high-pressure homogenization and sonication [48], or enzymatic pre-treatments [49,50]. Previous studies showed that incorporating undisrupted microalgae in animal diets may require double the amount to have the same effect as disrupted microalgae [11]. The polysaccharide content should also be considered, since high levels of polysaccharides can negatively affect protein digestibility [51].

Using in vitro experiments, Gerken et al. [49] tested the enzymatic (chitinase, lysozyme, pectinase, sulfatase, β-glucuronidase, and laminarinase) degradation of *Chlorella* cell walls and reported that this microalga was more sensitive to chitinase and lysozyme than to other enzymes. Both enzymes drastically affected cell permeability, thus influencing nutrient digestibility. According to Canelli et al. [50], enzymatic pre-treatments are a good choice for improvement of nutrient bioaccessibility, expressed as the proportion between the amount of nutrient incorporated into the micellar phase and that in full digesta. The bioaccessibility of proteins was particularly improved from 49.2 to 58.7%, without lipid oxidation and preserving cell wall integrity, opposing the mechanical pre-treatments, like high-pressure homogenization, which drastically affected oxidation and provoked some off-flavour formations although enhancing lipid accessibility (36.9 to 61.8%) compared to controls. Gille et al. [48], using an in vitro digestion model, concluded that sonication before digestion of CV, improved bioaccessibility of lutein at 11% (7 to 18% after sonication) and β-carotene at 12.5% (0 to 12.5% after sonication). Specifically, Kose et al. [52] tested the influence of pancreatin before in vitro protein digestion with trypsin and α-chymotrypsin, and apparent digestibility improved from 35% (protein digestibility of untreated CV) up to 70%. According to Wild et al. [53], the disruption of CV’s cell walls can enhance in vitro crude protein digestibility by 5% (79 to 84%), compared to non-cell-disrupted microalgae. An in vitro trial, where CV was digested with pepsin and pancreatin, reported some nutrient digestibility values. Digestibility was over 60% for dry matter, between 60 and 70% for carbohydrates and organic matter, and 76% for crude protein [14]. Table 2 summarises the main effects of in vitro pre-treatments on the hydrolysis and digestibility of CV.

A study by Neumann et al. [54] tested the impact of various pre-treatments on the digestibility of CV biomass in mice. The study incorporated 5, 15, and 25% of ball-milled CV (phototrophic or mixotrophic cultured) into mouse diets and found that protein availability was not impacted by the inclusion of CV up to 25%. However, the mixotrophically cultured CV at 25% had the lowest values of apparent digestibility (AD at 76.4%) and protein net utilization (NPU at 45.9%). Bead milling was identified as an effective method for disrupting the cell wall, thereby improving protein bioavailability. Moreover, the study determined the fatty acid content in livers and calculated the absorption index to assess the bioavailability of fatty acids. Despite the fact that CV-containing diets had ten times higher polyunsaturated fatty acids (PUFAs) than the control diet, the bioavailability of fatty acids was not affected and did not differ from control values.

Tsiplakou et al. [55] investigated the effect of 1% lyophilized CV on the chemical composition and fatty acid profile of goat’s milk and found no changes in the fatty acid profile. Meanwhile, Tibbetts et al. [56] studied the influence of cell-disrupted and non-disrupted CV on the diet of juvenile Atlantic salmon. The results showed that the inclusion of cell-disrupted *C. vulgaris* from 6 to 30% feed did not impact dietary dry matter digestibility, and similar results were obtained for lipid and protein with algal levels up to 18% and 24%, respectively. However, all incorporation levels improved the apparent digestibility coefficient of carbohydrates. The moderate inclusion of whole-cell CV up to 18% did not affect the dietary ADC for most essential amino acids, but high inclusion (24 to 30%) of cell-disrupted *C. vulgaris* did not affect this parameter for any essential amino acid.

In a trial conducted by Kholif et al. [57], the effect of dietary supplementation with 10 g/day of CV was evaluated in goats. The study involved two groups of goats, one that received the CV supplement along with copper (ALCU) and another without copper supplementation (AL). The results showed that the inclusion of CV improved the nutrient digestibility of crude protein, ether extract, neutral detergent fibre, and acid detergent fibre. Additionally, CV-containing diets led to an increase in the concentrations of total unsaturated fatty acids (9.8% and 5.4% for ALCU and AL, respectively), monounsaturated fatty acids (9.8% and 5.2% for ALCU and AL, respectively), and total conjugated linoleic acid (7.4% and 9.3% for ALCU and AL, respectively). The concentrations of saturated fatty acids also decreased by 4% for ALCU and 2.4% for AL, respectively. The improvement in nutrient digestibility was attributed to the presence of *Chlorella* growth factor [58]. Table 3 summarizes the main effects of different pre-treatments on the hydrolysis and digestibility of CV when tested in animal trials.

## 4. Impact of *Chlorella vulgaris* Biomass Digestibility in Poultry

The impact of CV on poultry digestion has been an area of interest for many years, dating back to 1950. Although Alshelmani et al. [59] refers to a widespread use of CV in poultry nutrition, there is still a lack of information on its effects on nutrient digestibility, bioavailability, and accessibility when incorporated into poultry diets.

The inclusion of CV in poultry diets has been studied for its potential to provide essential amino acids, fatty acids, and antioxidants. According to Kang et al. [60], 1% CV inclusion may impact the palatability of the diets, leading to reduced feed intake and average daily gain. On the other hand, Zheng et al. [61] found that incorporating 0.1 or 0.2% fermented CV in laying hen diets for 42 days improved egg production and yolk colour. The improvement was attributed to the enhanced availability of CV compounds after the fermentation process, which also positively impacted the hens’ digestive efficiency by altering the microflora profile in the ceca. The shift in microflora may have degraded algal polysaccharides and other components, contributing to more efficient digestion.

Alfaia et al. [38] recently evaluated the impact of 10% CV on broiler performance, meat quality, and lipid composition. The study found that the inclusion of CV, either alone or in combination with enzymes, led to an increase in the viscosity of the duodenum, jejunum, and ileum. The combination of CV with enzymes, Rovabio Excel AP (0.005%) and a mix of recombinant CAZymes (0.01%), resulted in higher viscosity compared to CV alone. Despite this, the study found that CV had a minor impact on the fatty acid composition in breast or thigh meat, but it did enhance some PUFAs, such as 18:3n-3, in the breast.

Roques et al. [62] also evaluated the effect of 0.8% dried powder CV in broiler diets on growth performance, immune response, and intestinal morphology. The study found that the inclusion of CV at this level had a positive impact on overall broiler performance and they maintained a strong immune response. These findings have led some animal nutritionists to consider the use of low doses of CV as a cost-effective alternative to traditional broiler feed formulations.

Kang et al. [63] and Mirzaie et al. [64] have studied the impact of *Chlorella* by-products on various aspects of poultry nutrition, such as broiler performance, meat quality, and gut health. These studies showed that incorporating 2.5, 5.0, or 7.5% *Chlorella* by-products into broiler diets increased villus height and crypt depth, which could enhance nutrient absorption and utilization [63]. Similarly, *Mirzaie* et al. [64] found that feeding 1 or 2% *Chlorella* by-products improved intestinal morphology by increasing villus height and crypt depth, and reducing the villus height to crypt depth ratio [64].

As described by Roques et al. [62] and Kang et al. [63], the small intestine, particularly the jejunum, plays a crucial role in digestive processes, such as enzyme digestion and nutrient uptake. The villus height reflects the surface area available for nutrient absorption, while the crypt depth indicates the rate of cell removal in the villi. A deeper crypt can suggest faster tissue turnover, which may be the organism’s response to counteract the effects of harmful toxins [62,63]. A decreased villus height to crypt depth ratio is an indication of improved digestive efficiency in the small intestine [63]. The development of favourable intestinal morphology is a hallmark of a healthy gut, with improved nutrient absorption and bioavailability for the animals [63]. Table 4 summarizes the effects of CV inclusion in poultry feeding.

Overall, CV is also a good source of essential amino acids, fatty acids, and antioxidants, and, therefore, its incorporation into poultry diets can enhance polyunsaturated fatty acids in meat, improve egg production, have positive effects in broiler performance and digestive efficiency, and induce a good immune response. However, this microalga might influence the viscosity of the duodenum, jejunum, and ileum when added at an ingredient level, which can compromise nutrient digestibility.

## 5. Influence of Chlorella vulgaris Biomass Digestibility in Swine

The impact of incorporating CV biomass into swine diets on bioaccessibility, bioavailability, and digestibility has been scarcely reported in the literature. Yan et al. [65] conducted a study involving the inclusion of 0.1 and 0.2% fermented CV in the diets of growing pigs. The results showed that the apparent total tract digestibility (ATTD) of nitrogen and energy was not affected, though there was a tendency for a slight decrease in ATTD of nitrogen (78.87 to 78.37%) and an almost 1% increase in ATTD of energy (75.74 to 76.94%) in comparison to the control group. However, the inclusion of 0.1% fermented microalga had a significant effect on dry matter ATTD, improving it from 76.04 to 78.61%. The authors also found that the inclusion of fermented CV reduced the concentration of E. coli and increased the concentration of Lactobacillus in the gut microbiome. This shift in microbial populations and decrease in faecal noxious gas content improved gut health and likely contributed to the increased ATTD [65,66]. The effects of incorporating this microalga on the gut microbiome of swine must be considered, as the intestine is a major site for nutrient absorption and plays a critical role in altering production performance [66]. According to Furbeyre et al. [67], the inclusion of 1% spray-dried Chlorella spp. in the diets of weaned piglets improved the ATTD of crude energy and showed a tendency to enhance the ATTD of dry matter, organic matter, and neutral detergent fibre. The height of the jejunum villus was also increased with Chlorella inclusion, thereby improving nutrient digestibility. During the crucial post-weaning period, Chlorella can effectively manage mild digestive problems, as demonstrated by a decrease in diarrhoea incidence with its inclusion. Conversely, Furbeyre et al. [68] reported shorter ileal villus and a higher and earlier occurrence of diarrhoea but with a fast recovery in piglets fed bead-milled Chlorella at 385 mg/kg body weight per day. This may be due to the bead-milling pre-treatment, which increased the viscosity and concentration of E. coli populations, leading to looser faeces.

The inclusion of 5% CV into the diets of finishing pigs was previously shown to improve lipid, antioxidant, pigment, and n-3 PUFA meat content, resulting in a reduction of the n-6:n-3 PUFA ratio and an overall improvement in the nutritional value of pork [35]. The combination of CV with a mixture of enzymes (0.005% Rovabio Excel AP or 0.01% mix of recombinant CAZymes) further enhanced C22:5n-3 and C22:6n-3 contents in meat by 1.6 times compared to control, without affecting microalgal digestive utilization by pigs under these conditions. Similarly, Martins et al. [15] found that incorporating CV, with or without enzymes, in piglet diets improved the nutritional value of meat by increasing the total carotenoid content (a two-fold increase compared to control) and n-3 PUFA while reducing the n-6:n-3 ratio. This positive result demonstrates a good correlation between the compounds found in microalgae and those deposited in muscle.

The study by Martins et al. [69] investigated the impact of 5% CV incorporation on nutrient digestibility of weaned piglets, either alone or in combination with enzymes (0.005% Rovabio Excel AP or 0.01% mix of recombinant CAZymes). The results showed that CV incorporation had a negative effect on ATTD, particularly of fibre, due to decreased effectiveness in CV cell wall disruption in the intestine. The viscosity of the duodenum and the height of the jejunum tended to increase with the addition of the microalga, but the simultaneous increase of duodenum villus height may have contributed to a healthier microbiota and improved gut health by stimulating prebiotic populations. The combination of CV and Rovabio resulted in values that were close to the control, suggesting a better degradation of the cell wall and improved nutrient digestibility [69].

Lastly, Ribeiro et al. [70] studied the impact of 5% CV incorporation, either alone or in combination with enzymes (0.005% Rovabio Excel AP or 0.01% mix of recombinant CAZymes), on the livers of finishing pigs. CV inclusion influenced lipid metabolism and oxidative stress, while the addition of CAZymes improved liver metabolism of n-3 PUFAs compared to the control group, leading to enhanced PUFA digestibility and hepatic metabolism. The combination of CV and CAZymes also decreased oxidative stress, which was suggested to be related to an increase in carotenoid content in the liver. The effects of CV inclusion in swine feeding are summarized in Table 5.

Overall, the incorporation of CV in swine diets may improve ATTD of gross energy and dry matter with a tendency to enhance the ATTD of nitrogen and organic matter, although it can negatively affect fibre ATTD. This microalga can also improve lipid metabolism, and, thus, increase n-3 PUFAs and decrease the n-6:n-3 PUFA ratio in meat.

## 6. Conclusions and Future Perspectives

This review showed that the impact of incorporating CV in poultry and swine diets varies and is influenced by multiple factors, including microalga strain, cultivation conditions, and dietary inclusion levels. However, pre-treatments applied to microalgal biomass can improve nutrient digestibility and accessibility. CV biomass can serve as a feed supplement or partial substitute for common feed sources, providing valuable basic nutrients, pigments, antioxidants, vitamins, growth factors, and prebiotics. This can increase the nutritional value of animal products, promote animal physiology and health, and ultimately lead to a more sustainable and profitable animal production system. Further research is required to optimize the application of CV in monogastric diets, including the selection of appropriate strains, cultivation conditions, pre-treatment methods, and inclusion level. This can lead to a better understanding of the effects of CV on animal health, digestion, and overall performance, and, thus, to a more widespread and efficient use of this microalga in animal nutrition. Additionally, further work is needed to investigate the mechanisms behind the positive effects of CV on animal health, including the role of microalgae on gut microbiota and regulation of oxidative stress. These insights will be critical in the development of more effective animal feeding strategies that enhance animal health, welfare, and performance.

## Figures and Tables

**Figure 1 animals-13-01017-f001:**
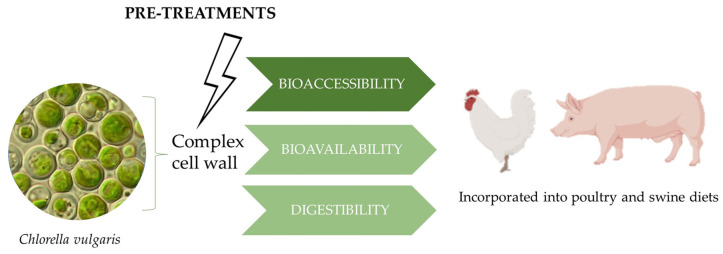
Bioaccessibility, bioavailability, and digestibility of Chlorella vulgaris fed to poultry and swine.

**Table 2 animals-13-01017-t002:** Summary of the main effects of in vitro pre-treatments on hydrolysis and digestibility of *Chlorella vulgaris* biomass.

Pre-Treatments	Main Effects	References
Six-enzyme mixture	Lysozyme and other enzymes drastically increased cell permeability	Gerken et al. [49]
Chitinase, rhamnohydrolase, and galactanase	Increased protein bioaccessibility (49.2 to 58.7% compared with control) and preserved cell integrity	Canelli et al. [50]
High-pressure homogenization	Increased lipid bioaccessibility (36.9 to 61.8% compared with control) and highly significant oxidative instability and development of off-flavours	Canelli et al. [50]
Sonication	Increased carotenoid bioaccessibility (10% for β-carotene and 15% for lutein)	Gille et al. [48]
Pancreatin	Apparent digestibility improved from 35 to 70%	Kose et al. [52]
Cell disruption with ball milling	Increased in vitro crude protein digestibility from 79 to 84%	Wild et al. [53]
Pepsin and pancreatin	Dry matter, carbohydrate and organic matter, and crude protein digestibility of 60%, 60–70%, and 76%, respectively	Niccolai et al. [14]

**Table 3 animals-13-01017-t003:** Summary of the main effects of pre-treatments combined with in vivo trials on hydrolysis and digestibility of *Chlorella vulgaris* (CV) biomass.

Pre-Treatments	Main Effects	References
5, 15, and 25% of ball-milled CV (phototrophic or mixotrophic cultured) in mice	No influence of microalga up to 25% feed on protein availability compared with control. Inclusion of CV mixotrophically at 25% led to the lowest values of ADC (76.4%) and net protein utilization (45.9%)	Neumann et al. [54]
1% lyophilized CV in goats	No impact on fatty acid profile	Tsiplakou et al. [55]
6, 12, 18, 24, and 30% of cell-disrupted CV (freeze-drying and high-pressure homogenization) in juvenile Atlantic salmon	Did not affect dry matter digestibility up to 30% of CV inclusion, protein up to 24%, lipid up to 18%, but improved carbohydrates ADC at all levels	Tibbetts et al. [56]
10 g/day of CV supplemented with copper in goats	Increased all nutrients digestibility. Increased concentrations of total unsaturated and monounsaturated fatty acids. Improved total conjugated linoleic acid	Kholif et al. [57]

CV: *Chlorella vulgaris;* ADC: apparent digestibility coefficient.

**Table 4 animals-13-01017-t004:** Summary of the main effects of inclusion of *Chlorella vulgaris* (CV) biomass or *Chlorella* by-products in poultry diets.

Animals (Age/Initial Body Weight)	Inclusion Level in Feed and Duration of Trial	Main Effects	References
1-day-old male broilers	1% CV for 4 weeks	Source of essential amino acids, fatty acids, and antioxidants. May affect palatability and reduce feed intake and daily gain	Kang et al. [60]
80-week-old laying hens	0.1 and 0.2% fermented CV for 42 days	Improved egg production and yolk colour. Positive impact on the animal’s digestive efficiency as it altered cecal microflora profile	Zheng et al. [61]
1-day-old male broilers	10% CV either alone or in combination with enzymes (0.005% Rovabio Excel AP and 0.01% mix of recombinant CAZymes) from 21 to 35 days	Increased viscosity in duodenum, jejunum, and ileum. Minor impact on fatty acid composition in breast or thigh meat but enhanced some polyunsaturated (*i.e.*, 18:3n-3 and 18:4n-3) fatty acids and decreased saturated fatty acids, mostly 16:0, in the breast	Alfaia et al. [38]
1-day-old male broilers	0.8% dried powder CV for 35 days	Positive impact on overall broiler performance and animals maintained a good immune response	Roques et al. [62]
1-day-old male broilers	2.5, 5.0, or 7.5% *Chlorella* by-products for 35 days	Increased villus height and crypt depth and decreased villus height to crypt depth ratio	Kang et al. [63]
22-day-old male broilers	1 or 2% *Chlorella* by-products for 21 days		Mirzaie et al. [64]

CV: Chlorella vulgaris.

**Table 5 animals-13-01017-t005:** Summary of the main effects of inclusion of Chlorella vulgaris (CV) biomass in swine diets.

Animals (Age/Initial Body Weight)	Inclusion Level in Feed and Duration of Trial	Main Effects	References
Growing pigs (26.58 ± 1.41 kg)	0.1 and 0.2% fermented CV for 6 weeks	Tendency to decrease nitrogen ATTD (78.87% to 78.37%). Increased energy ATTD by almost 1% (75.74 to 76.94%) and dry matter ATTD from 76.04% to 78.61% with 0.1% fermented microalga. Diminished concentration of *E. coli* population and enhanced concentration of *Lactobacillus.* Decreased faecal noxious gas content	Yan et al. [65]
28-day-old weaned piglets	1% spray-dried *Chlorella* from 28 to 42 days	Improved ATTD of gross energy and tended to enhance dry matter, organic matter, and neutral detergent fibre ATTD. Increased the height of jejunum villi and decreased diarrhoea	Furbeyre et al. [67]
28-day-old weaned piglets at	Oral supplementation of bead-milled *Chlorella* (385 mg/kg body weight per day) for 4 weeks	Shorter ileal villi and an earlier and higher occurrence of diarrhoea but with faster recuperation	Furbeyre et al. [68]
Finishing pigs (59.1 ± 5.69 kg)	5% CV either alone or in combination with enzymes (0.005% Rovabio Excel AP and 0.01% mix of recombinant CAZymes) for 41 ± 7.8 days	Increased antioxidants, pigments, and n-3 PUFAs and reduced n-6:n-3 PUFA ratio in pork	Coelho et al. [35]
Post-weaned male piglets (11.2 ± 0.46 kg)	5% CV either alone or in combination with enzymes (0.005% Rovabio Excel AP and 0.01% mix of recombinant CAZymes) for 2 weeks		Martins et al. [15]
28-day-old weaned piglets	5% CV either alone or in combination with enzymes (0.005% Rovabio Excel AP and 0.01% mix of recombinant CAZymes) for 21 days	ATTD was negatively affected by CV inclusion, principally that of fibre. Tendency to increase viscosity of duodenum and jejunum height and enhancement of duodenum villus height	Martins et al. [69]
Finishing pigs (59.1 ± 5.69 kg)	5% CV either alone or in combination with enzymes (0.005% Rovabio Excel AP and 0.01% mix of recombinant CAZymes) until animal reached 101 ± 1.9 kg	Affected lipid metabolism and oxidative stress. Incorporation of CAZymes increased liver metabolism of n-3 PUFAs and decreased oxidative stress. Enhancement of PUFA digestibility and hepatic metabolism	Ribeiro et al. [70]

CV: *Chlorella vulgaris*; ATTD: apparent total tract digestibility; PUFA: polyunsaturated fatty acid.

## Data Availability

The data presented in this study are available on request from the corresponding author.

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
