# Peer review of "Enhancing Digestibility of Chlorella vulgaris Biomass in Monogastric Diets: Strategies and Insights"

_animals, 2023, doi:10.3390/ani13061017_

Round 1

Reviewer 1 Report

This review suffers some major fails in the disposition, terminology, writing style, readability and fluency. Nonetheless, the manuscript could be much improved and could be a useful contribution to our knowledge on the subject and can be accepted after these minor revisions:

1- English should improve by a native person. The paper suffers from a poor English structure throughout and cannot be published or reviewed properly in the current format. The manuscript requires a thorough proofread by a native person whose first language is English. The instances of the problem are numerous and this reviewer cannot individually mention them. It is the responsibility of the author(s) to present their work in an acceptable format. Unless the paper is in a reasonable format, it should not have been submitted.

2- The novelty of the review needs to be highlighted compare to other similar reviews or consider to explicitly mention what is gap knowledge and/or what was lacking in the indicated studies.

3- The scientific background of the topic is poor. The authors should cite recent references between 2018-2022 from JCR journals (with impact factor) about recent achievements on the algae feeding in commercial poultry. For example, authors should cite to:

Coudert, E., Baéza, E., & Berri, C. (2020). Use of algae in poultry production: A review. World's Poultry Science Journal, 76(4), 767-786.

Feshanghchi, M., Baghban-Kanani, P., Kashefi-Motlagh, B., Adib, F., Azimi-Youvalari, S., Hosseintabar-Ghasemabad, B., Slozhenkina, M., Gorlov, I., Zangeronimo, M.G., Swelum, A.A., Seidavi, A. R., Khan, R. U., Ragni, M., Laudadio, V. and Tufarelli, V. 2022. Milk Thistle (Silybum marianum), Marine Algae (Spirulina platensis) and Toxin Binder Powders in the Diets of Broiler Chickens Exposed to Aflatoxin-B1: Growth Performance, Humoral Immune Response and Cecal Microbiota. Agriculture-Basel. 12(6): 805, 1-11.

4- A detailed "Conclusion" should be provided to state the final review output that the authors have reached. Please note you only need to place your conclusion and not keep putting other results, because these have already been presented in the manuscript.

5- Author(s) should re-format the references based on journal format. See the instructions for authors.

6- A schematic figure could improve the aspect and serve for a better visualization of the review. 

Author Response

Reviewer 1

This review suffers some major fails in the disposition, terminology, writing style, readability and fluency. Nonetheless, the manuscript could be much improved and could be a useful contribution to our knowledge on the subject and can be accepted after these minor revisions:

  • English should improve by a native person. The paper suffers from a poor English structure throughout and cannot be published or reviewed properly in the current format. The manuscript requires a thorough proofread by a native person whose first language is English. The instances of the problem are numerous and this reviewer cannot individually mention them. It is the responsibility of the author(s) to present their work in an acceptable format. Unless the paper is in a reasonable format, it should not have been submitted.

Reply: Thank you for your observation. English was improved throughout the manuscript and revised by a native English speaker, Dr. Shabir Najmudin.

2- The novelty of the review needs to be highlighted compare to other similar reviews or consider to explicitly mention what is gap knowledge and/or what was lacking in the indicated studies.

Reply: Thank you for your comment. This review not only summarizes in vivo studies where Chlorella vulgaris was incorporated in poultry and swine diets, but also focus on in vitro studies and highlights the importance of pre-treatments in improving digestibility and bioaccessibility microalga nutrients. This aspect is now specified between lines 82 and 96, page 2.

3- The scientific background of the topic is poor. The authors should cite recent references between 2018-2022 from JCR journals (with impact factor) about recent achievements on the algae feeding in commercial poultry. For example, authors should cite to:

Coudert, E., Baéza, E., & Berri, C. (2020). Use of algae in poultry production: A review. World's Poultry Science Journal76(4), 767-786.

Feshanghchi, M., Baghban-Kanani, P., Kashefi-Motlagh, B., Adib, F., Azimi-Youvalari, S., Hosseintabar-Ghasemabad, B., Slozhenkina, M., Gorlov, I., Zangeronimo, M.G., Swelum, A.A., Seidavi, A. R., Khan, R. U., Ragni, M., Laudadio, V. and Tufarelli, V. 2022. Milk Thistle (Silybum marianum), Marine Algae (Spirulina platensis) and Toxin Binder Powders in the Diets of Broiler Chickens Exposed to Aflatoxin-B1: Growth Performance, Humoral Immune Response and Cecal Microbiota. Agriculture-Basel. 12(6): 805, 1-11.

Reply: We acknowledged your comments and suggestions and cited the references proposed (lines 46 and 50 to 55, page 2). The majority of references used in this review are from 2017 to 2022.

4- A detailed "Conclusion" should be provided to state the final review output that the authors have reached. Please note you only need to place your conclusion and not keep putting other results, because these have already been presented in the manuscript.

Reply: Thank you for your comment. We changed the conclusion accordingly.

5- Author(s) should re-format the references based on journal format. See the instructions for authors.

Reply: Thank you for your comment and suggestion. The references were checked and some were changed according to the journal format.

6- A schematic figure could improve the aspect and serve for a better visualization of the review.

Reply: Thank you for your comment. Figure 1 was now added to the manuscript (lines 96-98, page 3).

Reviewer 2 Report

The paper contains valuable data. Results were properly reported, and the findings have been accurately discussed and compared with other published papers. For further improvement of the manuscript, it requires some modification.

P1,L13 = Simple Summary

Change “The results of a systematic review of literature showed that incorporating CV in poultry and swine diets had varying effects, but pretreatments improved nutrient digestibility and accessibility.” to be “According to the findings of a systematic review of the literature, adding CV to the diets of pigs and poultry had different results, although pretreatments increased nutrient accessibility and digestibility.”.

P1,L26 = Abstract

Change “Results indicated that incorporating CV in poultry and swine diets showed varying effects, which are influenced by several factors.” to be “The results demonstrated that adding CV to the diets of pigs and poultry had diverse results, depending on a number of variables.”.

P1,L37 = Introduction

Please indicate the reasons for using this type of feed in livestock and poultry feed.

You can use new references such as:

Nobakht, A., Palangi, V., AyaÅŸan, T., & Coçkun, I. (2022). Efficacy of Tragopogon graminifolius medicinal powder as an inulin source for laying hens. South African Journal of Animal Science, 52(3), 252-258.

Palangi, V., & Lackner, M. (2022). Management of Enteric Methane Emissions in Ruminants Using Feed Additives: A Review. Animals, 12(24), 3452.

P3,L124 = Table 1.

It is better to convert this table into a figure.

P11,L321 = Conclusions and Future Perspectives

Discussion explained adequately.

P12,L353 = References

References are adequate.

Regards

Author Response

Reviewer 2

The paper contains valuable data. Results were properly reported, and the findings have been accurately discussed and compared with other published papers. For further improvement of the manuscript, it requires some modification.

Reply: Thank you for your comments and suggestions. We appreciate it. We tried to address all of them.

P1,L13 = Simple Summary

Change “The results of a systematic review of literature showed that incorporating CV in poultry and swine diets had varying effects, but pre-treatments improved nutrient digestibility and accessibility.” to be “According to the findings of a systematic review of the literature, adding CV to the diets of pigs and poultry had different results, although pre-treatments increased nutrient accessibility and digestibility.”

Reply: Thank you for your suggestion. The sentence was changed to: ”The findings of the present systematic review show that adding CV to poultry and swine diets had different results in terms of nutrient digestibility, although pre-treatments increased nutrient accessibility and digestibility”.

P1,L26 = Abstract

Change “Results indicated that incorporating CV in poultry and swine diets showed varying effects, which are influenced by several factors.” to be “The results demonstrated that adding CV to the diets of pigs and poultry had diverse results, depending on a number of variables.”.

Reply: Thank you for your suggestion. The sentence was changed to: “The results of adding CV to poultry and swine diets were diverse and depended of a number of variables”

P1,L37 = Introduction

Please indicate the reasons for using this type of feed in livestock and poultry feed.

You can use new references such as:

Nobakht, A., Palangi, V., AyaÅŸan, T., & Coçkun, I. (2022). Efficacy of Tragopogon graminifolius medicinal powder as an inulin source for laying hens. South African Journal of Animal Science, 52(3), 252-258.

Palangi, V., & Lackner, M. (2022). Management of Enteric Methane Emissions in Ruminants Using Feed Additives: A Review. Animals, 12(24), 3452.

Reply: Thank you for your suggestion. The reasons for using microalgae in animal feed are now presented in the Introduction section, page 2, lines 50-55, as well as the citation (Palangi et al., 2022). The reference Nobakht et al. (2022) was not added because it describes the effects of plant-derived compounds.

P3,L124 = Table 1.

It is better to convert this table into a figure.

Reply: Thank you for your suggestion. However, we cannot convert the table into a figure due to high number of variables present in the table, which was necessary to gather the maximum information available in the literature.  

P11,L321 = Conclusions and Future Perspectives

Discussion explained adequately.

Reply: Thank you for your comment. We appreciate it.

P12,L353 = References

References are adequate. 

Reply: Thank you for your comment. We appreciate it.

Reviewer 3 Report

Dear authors, this review provides new and important data about the use of microalga Chlorella vulgaris (CV) as an animal feed source. Moreover, the use of English language in the present study was appropriate and only some phrases and sections such as simple summary must be rewritten. However, authors must change and delete several parts of the manuscript in order to provide more accurate conclusions. Conclusively this work needs to be minor revised to be published in this journal. The questions which must be answered and the changes which are proposed to be done, are presented line by line in the following paragraphs.

Simple summary:

L. 12-19. Please rewrite the simple summary because the reader remains with the impression that this study it is not a review research but an in vivo experiment.

1. Introduction of Chlorella vulgaris

---------------------------------------------------------

2. Nutritional Composition of Chlorella vulgaris

---------------------------------------------------------

3. Enhancing the Digestibility of Chlorella vulgaris Nutrients

L. 134-140. It is not necessary to provide the usage of nutrient digestibility measurements. The authors must delete this paragraph.

4. Impact of Chlorella vulgaris Biomass Digestibility in Poultry

L. 213-214. According to the fact that this review refers to recent studies until January 2023, it is not appropriate to mention references back from 1950. Moreover, authors must provide a conclusion in each section based on the results of the majority of the examined studies.

5. Influence of Chlorella vulgaris Biomass Digestibility in Swine

The same thing as in section 4, authors must provide a conclusion in each section based on the results of the majority of the examined studies.

Conclusions:

----------------------------------------------------------

Author Response

Reviewer 3

Dear authors, this review provides new and important data about the use of microalga Chlorella vulgaris (CV) as an animal feed source. Moreover, the use of English language in the present study was appropriate and only some phrases and sections such as simple summary must be rewritten. However, authors must change and delete several parts of the manuscript in order to provide more accurate conclusions. Conclusively this work needs to be minor revised to be published in this journal. The questions which must be answered and the changes which are proposed to be done, are presented line by line in the following paragraphs.

Reply: Thank you for your comments and suggestions. We appreciate it. We tried to address all of them.

Simple summary:

  1. 12-19. Please rewrite the simple summary because the reader remains with the impression that this study it is not a review research but an in vivo experiment.

Reply: Thank you for your suggestion. The simple summary in know rewritten.

  1. Introduction of Chlorella vulgaris

---------------------------------------------------------

  1. Nutritional Composition of Chlorella vulgaris

---------------------------------------------------------

  1. Enhancing the Digestibility of Chlorella vulgaris Nutrients
  2. 134-140. It is not necessary to provide the usage of nutrient digestibility measurements. The authors must delete this paragraph.

Reply: Thank you for your suggestion. The paragraph was deleted.

  1. Impact of Chlorella vulgaris Biomass Digestibility in Poultry
  2. 213-214. According to the fact that this review refers to recent studies until January 2023, it is not appropriate to mention references back from 1950. Moreover, authors must provide a conclusion in each section based on the results of the majority of the examined studies.

Reply: Thank you for your suggestion. We used the references back from 1950 to prove that incorporation of Chlorella vulgaris in poultry diets has been commonly used through many years. But we took your comment into consideration and we removed the corresponding citation.

Also, the main conclusions of the results of most of the examined studies in poultry are now summarized after corresponded section (lines 295 to 302, page 9).

  1. Influence of Chlorella vulgaris Biomass Digestibility in Swine

The same thing as in section 4, authors must provide a conclusion in each section based on the results of the majority of the examined studies.

Reply: Thank you for your suggestion. The same thing as in section 4 was done for section 5 (lines 362 to 366, page 11).

Conclusions:

----------------------------------------------------------